# Acceptability of risk-based triage in cervical cancer screening: A focus group study

Sharell Bas [1], Jasmijn Sijben [2], Erik W. M. A. Bischoff[3], Ruud L. M. Bekkers [4,5], Inge M. C. M. de Kok[6], Willem J. G. Melchers[7], Albert G. Siebers [8], Daniëlle van der Waal[1,9], Mireille J. M. Broeders [1,9]*

1 Radboud University Medical Center, Radboud Institute for Health Sciences, Nijmegen, The Netherlands, 2 Department of Gastroenterology and Hepatology, Radboud Institute for Health Sciences, Radboud University Medical Center, Nijmegen, The Netherlands, 3 Department of Primary and Community Care, Radboud Institute for Health Sciences, Radboud University Medical Center, Nijmegen, Netherlands, 4 Department of Obstetrics and Gynaecology, Catharina Hospital, Eindhoven, The Netherlands, 5 Department of Obstetrics and Gynecology, GROW School for Oncology and Reproduction, Maastricht University Medical Center, Maastricht, The Netherlands, 6 Department of Public Health, Erasmus MC University Medical Center, Rotterdam, The Netherlands, 7 Department of Medical Microbiology, Radboud University Medical Centre, Nijmegen, The Netherlands, 8 The Nationwide Network and Registry of Histo-and Cytopathology in the Netherlands (PALGA Foundation), Houten, The Netherlands, 9 Dutch Expert Centre for Screening, Nijmegen, The Netherlands

* Mireille.Broeders@radboudumc.nl

**Data Availability Statement:** We have serious ethical concerns about making the transcripts available in a public repository. The size of the groups in combination with the setting (a limited

## Abstract

### Background

Compared to the previous cytology-based program, the introduction of primary high-risk human papillomavirus (hrHPV) based screening in 2017 has led to an increased number of referrals. To counter this, triage of hrHPV-positive women in cervical cancer screening can potentially be optimized by taking sociodemographic and lifestyle risk factors for cervical abnormalities into account. Therefore, it is essential to gain knowledge of the views of women (30–60 years) eligible for cervical cancer screening.

### Objective

The main goal of this qualitative study was to gain insight in the aspects that influence acceptability of risk-based triage in cervical cancer screening.

### Design

A focus group study in which participants were recruited via four general medical practices, and purposive sampling was used to maximize heterogeneity with regards to age, education level, and cervical cancer screening experiences.

### Approach

The focus group discussions were transcribed verbatim and analyzed using reflexive thematic analysis.

number of GP practices) makes it impossible to guarantee that participants cannot be identified through the transcripts. We also did not ask for our participants' consent to make the transcripts available publicly. This was a choice based on the sensitive nature of the topics described in the focus groups. Other researchers who want to replicate the study can reach out to the ethics board of the Radboudumc, Nijmegen, for consideration via email (METCoost-en-CMO@radboudumc.nl).

**Funding:** This study is an independent study funded by The Netherlands Organization for Health Research and Development (ZonMw, https://www.zonmw.nl/en/, project no. 555004204). The funders had no role in study design, data collection and analysis, decision to publish, or preparation of the manuscript.

**Competing interests:** The authors have declared that no competing interests exist.

## Participants

A total of 28 women (average age: 45.2 years) eligible for cervical cancer screening in The Netherlands participated in seven online focus group discussions. Half of the participants was higher educated, and the participants differed in previous cervical cancer screening participation and screening result.

## Key results

In total, 5 main themes and 17 subthemes were identified that determine the acceptability of risk-stratified triage. The main themes are: 1) adequacy of the screening program: an evidence-based program that is able to minimize cancer incidence and reduce unnecessary referrals; 2) personal information (e.g., sensitive topics and stigma); 3) emotional impact: fear and reassurance; 4) communication (e.g., transparency); and 5) autonomy (e.g., prevention).

## Conclusion

The current study highlights several challenges regarding the development and implementation of risk-based triage that need attention in order to be accepted by the target group. These challenges include dealing with sensitive topics and a transparent communication strategy.

## Introduction

Cervical cancer population screening programs have successfully reduced cervical cancer incidence and mortality [1, 2]. In 2017, high-risk human papillomavirus (hrHPV) screening was introduced as the primary screening test in the Dutch cervical cancer screening program, for all women aged 30–60 years. After a hrHPV-negative result, no further testing is required. HrHPV-positive women are invited for a check-up smear after six months when triage cytology is normal (Papanicolaou 1; pap1), or referred to the gynecologist for further testing when any cytological abnormality is encountered (pap2+). Compared to the previous cytology-based program, the introduction of primary hrHPV-based screening has led to a higher sensitivity but also to a lower specificity [3]. As a consequence, the detection rate for cervical intraepithelial neoplasia grade 2 or worse (CIN2+) increased from 999 to 1461 per 100,000 among women screened between 2014/2015 and 2017/2018. On the other hand, clinically irrelevant findings (≤CIN1) increased from 611 to 1487 per 100,000 among women screened [4]. Women experience a high level of anxiety when found to be hrHPV-positive [5] and being referred for a colposcopy [6, 7]. To reduce unnecessary referrals of lesions that are not clinically relevant (≤CIN1), triage of hrHPV-positive women has been, since July 2022, based on both hrHPV genotyping and cytology, which can be determined from one and the same smear taken by the general practitioner [8, 9].

A number of sociodemographic and lifestyle risk factors for CIN2+ are known, including smoking behavior [10], oral contraceptive use [11], parity [12], immunosuppressant use [13], and number of lifetime sexual partners [14, 15]. These factors are either associated with getting infected with hrHPV, progression to CIN2+ in women who are already hrHPV-positive, or both. Some studies indicate that risk prediction models can identify individuals who have a

higher risk of developing CIN2+ based on these sociodemographic and lifestyle risk factors [16–18]. In an optimized risk-based scenario, the individual risk of developing CIN2+ would determine the intensity of the follow-up trajectory, improving the population benefit-harm ratio. A recent study has indeed shown that accurate risk classification can be improved to develop a personalized cervical cancer screening program [19].

Before further investigating the possibilities of risk-based triage in the cervical cancer screening program, it is essential to gain insights into the views of the women eligible for cervical cancer screening. For example, because some risk factors for cervical abnormalities may be sensitive, such as questions related to lifestyle and sexual behavior, it is important to investigate the willingness among women to share such information in a questionnaire. Moreover, risk profiling implies that people are categorized into a risk category. It is therefore also important to explore women's views on being categorized based on cervical cancer risk. Since few studies on this topic are available, qualitative research can provide novel insights by capturing experiences and inform further quantitative research [20].

The aim of the current focus group study was to gain more insight into the acceptability of risk-based triage based on self-reported information on sociodemographic and lifestyle characteristics among women who are eligible for the Dutch cervical cancer screening program.

## Materials and methods

### Study design and ethics approval

Focus group discussions (FGDs) were performed to explore women's perceptions of risk-based triage in cervical cancer screening. All participants provided written informed consent. According to the Medical Ethics Review Committee of Radboudumc, Nijmegen, the study did not fall within the remit of the Dutch Medical Research Involving Human Subjects Act (WMO) and could be carried out in the Netherlands without ethical approval from an ethics committee (2021.8094). We followed the Consolidated Criteria for Reporting Qualitative Research (COREQ, S1 Table) [21].

### Recruitment and sampling

This focus group study aimed to include women in the target group of the Dutch cervical cancer screening program, i.e., women between 30 and 60 years old. We developed an information letter about the study and a leaflet about privacy and data storage. These information materials were discussed with a panel from Stichting ABC, a Dutch organization by and for lower-literate individuals, to improve comprehension and facilitate decision-making. An e-mail which provided brief information about the study and the researchers' contact details, was sent to all female patients aged 30 to 60 years old who were registered in four general practices. The general practices were selected because they are core practices in the Radboudumc General Practice Network and are familiar with participating in scientific research. The practices are located in different neighborhoods in the city of Nijmegen and cover a broad range of socioeconomic status and ethnic backgrounds. General practitioners could exclude patients from the e-mail list for medical reasons, such as terminally ill or severe mental disorders. In total, 5904 women were approached in May and June 2021. All women who participated in the study, had received the information letter and privacy leaflet prior to the FGD. Because the FGDs were in Dutch, only Dutch-speaking women could take part in the study.

Women who expressed being interested in participating ($n = 110$) were sent a questionnaire with the following questions: 1) age; 2) education level; 3) ever participated in the regular cervical cancer screening program; 4) ever tested positive for hrHPV; 5) ever had a check-up smear, and if yes, in which year; 6) ever had a referral to the gynecologist, and if yes, in which year.

Education level was dichotomized consistent with the classification of Statistics Netherlands [22]. A higher education level was defined as having completed university or university of applied sciences, and a lower-intermediate education level includes primary education, prevocational secondary education, senior general secondary education, pre-university education, and senior secondary vocational education. Using purposive sampling based on the questionnaire answers, we aimed to maximize the heterogeneity of the women participating in the FGDs with regards to age, education level and cervical cancer screening experiences.

## Topic guide

The FGDs started with a short presentation about the current Dutch cervical cancer screening program including the triage of hrHPV-positive women, and the goals of our study. We followed a semi-structured topic guide with open-ended questions (S2 Table). First of all, we explored attitudes to implementing a risk-based triage strategy in the cervical cancer screening program in the Netherlands, using questions and possible (triage) scenarios to facilitate the discussion. Moreover, for several potential risk factors for cervical cancer, we asked about willingness to provide information about these factors. Furthermore, questions were asked about information needs and stigma regarding risk-based triage in cervical cancer screening. After the first FGD, minor changes were made to the topic guide.

## Procedure

The FGDs were planned using an online calendar tool and were carried out in June and July 2021. Reminder emails were sent. All FGDs were organized and moderated by the first author (SB), a female researcher (MSc) who is trained and experienced in conducting qualitative research. The researchers were experienced in (risk-based) cancer screening research and did not disclose any further details about their personal goals and interests. Seven FGDs were held with 28 participants in total (2–7 participants per FGD). Because of covid-19 regulations, all FGDs were held online via Zoom. Most participants were at home in a room with no other people present. In the largest FGD, engagement varied because of several reasons, such as distractions at home. However, the (partial) absence of some participants did not negatively impact the active discussion between the other participants. In the other FGDs, on the other hand, we did not experience any negative effects of conducting an online discussion compared to a face-to-face discussion. The FGDs had an average duration of 80 minutes (60–105 minutes) and were audio-recorded. We made notes during and directly after the FGDs to remember topics to be discussed later and to facilitate interpretation of the results. We continued until it we had the impression that the FGDs provided little new information. No repeat interviews were carried out. Any identifiers, including names, were removed from the research data and stored securely.

## Data analysis

The FGDs were transcribed verbatim. Information that could lead to identification of participants, such as name or place of residence, was not included in the transcripts. The transcripts were not returned to the participants. The FGD transcripts were analyzed inductively using reflexive thematic analysis [23–25]. In reflexive thematic analysis, the researchers have an active and creative role. This method can produce rich interpretations of the patterns in the data [25] and is a six-step iterative process: 1) data familiarization; 2) systematic data coding; 3) generating initial themes; 4) developing and reviewing themes; 5) reviewing, defining and naming themes; and 6) writing the report. All transcripts were analyzed independently in pairs of two researchers using Atlas.ti 22 (step 1–4): SB analyzed all transcripts, and DvdW and JS

analyzed half of the transcripts each. In several discussions, SB, DvdW, and JS explored different perspectives on the data and further developed, defined, named, and, where appropriate renamed, the themes and subthemes (step 4–5). The participants did not analyze the data or provide feedback on the findings.

## Results

### Participants

Almost 6000 women received the study information e-mail from their general practitioner. In total, 101 women completed the questionnaire. We invited 42 women using purposive sampling and 28 of them participated (Fig 1).

The average age of the participants ($n$ = 28) was 45.2 years ($SD$ = 7.6), and half of the participants was higher educated (Table 1, see also S3 Table for characteristics per participant). The participants differed in cervical cancer screening participation and in whether they had ever been invited for follow-up testing.

### Factors that influence acceptability of risk-profiling

In total, five main themes and seventeen subthemes were constructed (Fig 2). The themes are: 1) adequate screening program; 2) personal information; 3) emotional impact; 4) communication; and 5) autonomy. Example quotes are provided in S4 Table.

**Adequate screening program.**   An adequate screening program is the most important factor for the acceptability of risk-based triage in cervical cancer screening. First of all, women stressed the importance of evidence-based risk profiling, making use of reliable models that include well-defined risk factors. Second, the importance of a high detection rate of (pre-) cancerous lesions was mentioned often. Women appreciated a more intensive follow-up for women who are classified as having a higher risk of cervical cancer. They expect that the additional time and attention dedicated to this group, will lead to less invasive treatments and a reduced mortality. However, it was pointed out that a potential new triage strategy should not cause a higher number of missed cases of cervical cancer in women at lower risk:

> ""*[it is important to me] that for me as a person the chance that I do have it and that it will not be picked up does not increase. That would be relevant to me." (FGD3, participant 8, aged 46)*

Third, the risk classification should be accurate, i.e., an individual's risk score or risk group should correspond to their actual risk. It should be anticipated that some women might not know the answer to a risk factor question or are not willing to provide it. It should also be taken into account that situations regarding risk factors (e.g., smoking, marital status) may change over time:

> "*Who's to say she neatly sticks with her own partner? And not get a divorce next year, which might put her at high risk again?" (FGD4, participant 15, aged 44)*

Fourth, trust influenced the acceptability of risk profiling. Some women mentioned they would trust a potential new triage strategy. However, some women would trust the current strategy more, because they are already used to this. Fifth, women were, in general, positive about less unnecessary follow-up testing for women who are at a lower risk of developing cervical cancer:

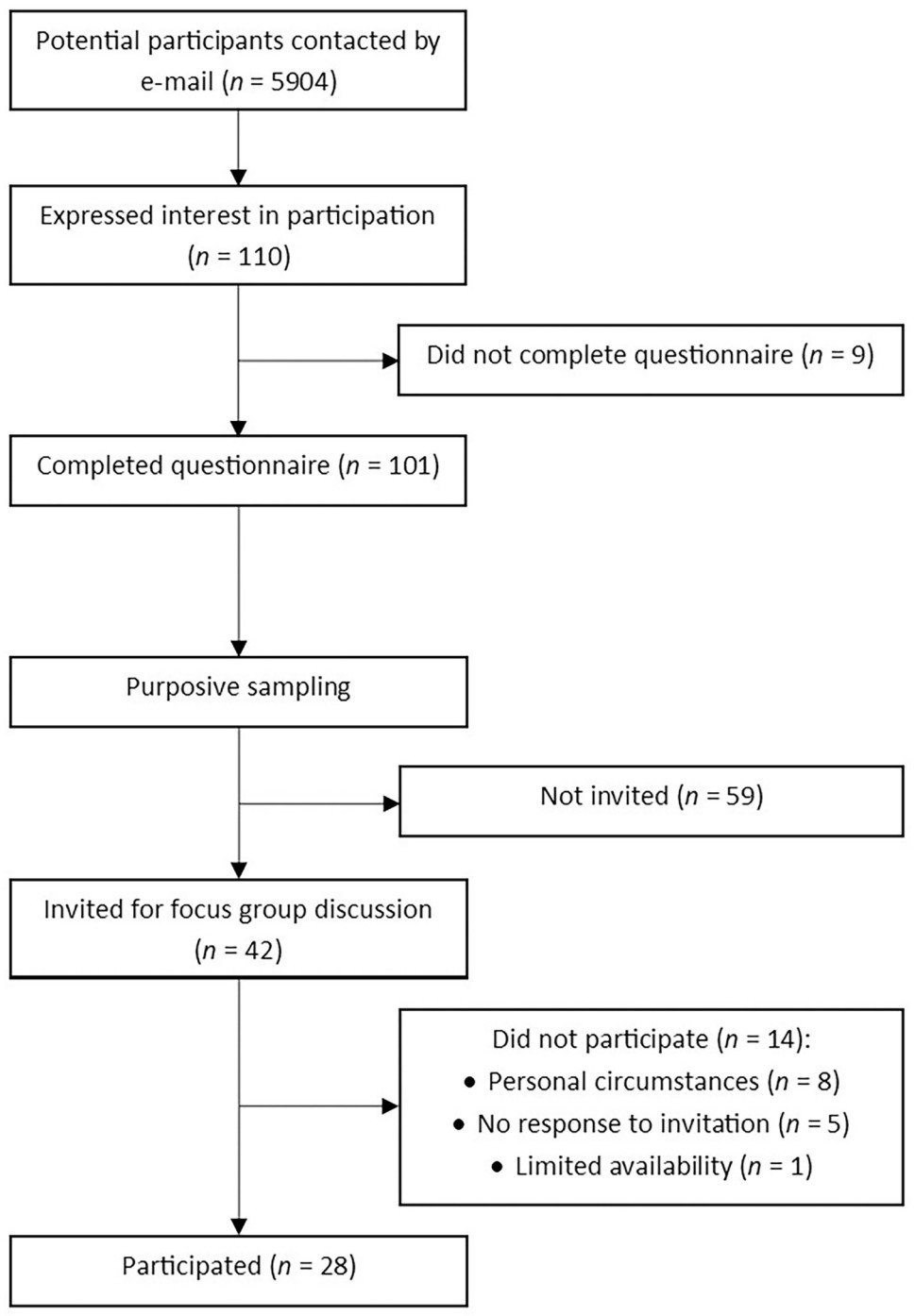

**Fig 1. Flowchart of participant recruitment.**

> "*Because of course it is also very nice for the patient herself that they do not have to go through a whole medical process that may not lead to anything at all.*" (FGD5, participant 17, aged 42)

**Personal information.** First, intrusiveness was an important factor to accept risk-based triage. Some questions, especially questions about sexual history, were experienced as sensitive,

**Table 1. Demographic and cervical cancer screening characteristics of the study sample.**

| Characteristics | N |
|---|---|
| Education level* | |
| Higher | 14 |
| Lower-intermediate | 14 |
| Ever participated in cervical cancer screening? | |
| Yes | 21 |
| No | 7 |
| Ever tested positive for HPV?† | |
| Yes | 6 |
| No | 12 |
| Don't know | 3 |
| Ever had a check-up smear or a referral to the gynecologist?† | |
| Yes | 10 |
| 2017 or later (current program) | 4 |
| No | 11 |

\* Consistent with the classification of Statistics Netherlands, education level was dichotomized into higher (completed university or university of applied sciences) and lower-intermediate (other education levels).

† Only women who had ever participated in cervical cancer screening were asked this question.

personal, intimate, or confronting. Second, women had mixed expectations regarding stigma of having a high risk of cervical cancer. On the one hand, some women mentioned that some high-risk women may fear stigma, because people may associate HPV positivity and high cervical cancer risk with sexual behavior. However, the likelihood of facing stigma could be reduced by deliberately selecting the people to share the information with. On the other hand, some women did not consider stigma an important issue, and indicated that they would not judge high-risk women negatively:

> "If a friend were to say that to me, my first reaction would just be: 'how awful'. But I wouldn't think about it like: 'why is she at higher risk?' No, I don't think so." (FGD3, participant 8, aged 46)

Understanding the relevance of sensitive questions was the third subtheme. Women expect that information about why a question is asked, would promote completing the questionnaire and thus increase the support for risk-based triage:

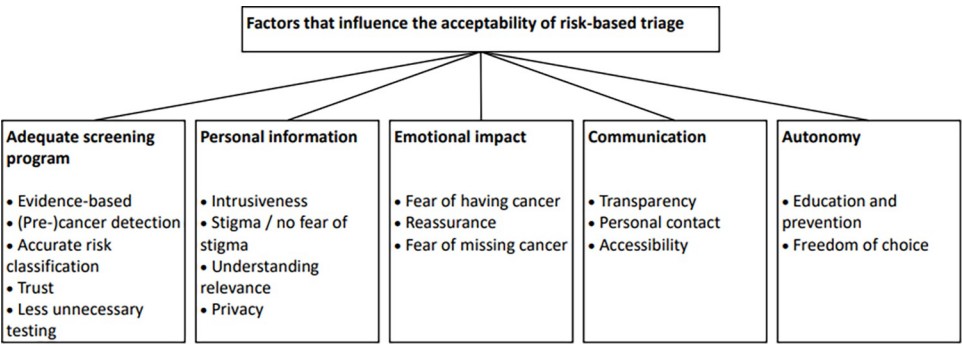

**Fig 2. Themes and subthemes in acceptability of risk profiling in cervical cancer screening.**

"*Why do I have to answer this? What's in it for me? And when you know that, I think a lot of barriers are removed.*" *(FGD4, participant 13, aged 46)*

Lastly, given the sensitive nature of some of the proposed risk factor questions, privacy was considered essential. To give an example, women suggested to add a "don't want to answer" option to certain questionnaire items. This point was mainly raised in relation to the sexual history risk factors. To give another example, it needs to be clear that the information is stored securely.

**Emotional impact.** A cervical cancer screening program can emotionally impact women in different ways. First, some aspects of cervical cancer screening, such as waiting time or a high-risk result, can cause a fear of having cancer. In this respect, a reduced number of unnecessary referrals and check-up smears can be perceived as desirable, which is a view that was often supported by personal experiences:

"*I sometimes have to wait for check-up results and that just causes a lot of stress. And knowing that you have a low risk is more pleasant than thinking: 'I have to come back, because I have a very high risk', while it turns out not to be the case.*" *(FGD1, participant 2, aged 56)*

The second subtheme within this theme, is reassurance. The communication strategy around risk profiling in the triage of cervical cancer screening should offer reassurance, for example by illustrating how common HPV is and that the risk of getting cervical cancer is relatively low. Moreover, it is recommended to avoid the word "risk", as this word may be perceived as scary and threatening. Third, women also expressed a fear of missing cancer in a potential new screening program that includes risk-based triage, especially in low-risk individuals who may be followed-up less intensively:

"*The question is: do you feel fooled if you do eventually get sick?*" *(FGD2, participant 5, aged 54)*

**Communication.** First of all, transparency was considered most important concerning the communication about the general cervical cancer screening program and the triage strategy in particular. For example, women appreciated acknowledgement that no program is able to offer 100% certainty. To give another example, clear and credible information about personal risk and follow-up was seen as crucial. Second, although a minority of the women preferred a low risk result to be communicated by a letter, most women expressed a wish for personal contact when receiving the result, regardless of the result (classified as low or high risk):

"*I like an in-person meeting better, because a letter is so impersonal. If I get questions during such a conversation, I can immediately ask them, because then, I think, I will also be reassured.*" *(FGD5, participant 18, aged 45)*

Third, the importance of accessible communication around risk profiling in the triage of cervical cancer screening was stressed. The risk factor questions should be clear, and communication should be suited for a large group of people, including people of all language and education levels. One example that was mentioned multiple times, is the addition of visual information to support the text:

"*I think that a combination with pictures—you can make it a bit visual—makes it more accessible for many people.*"*(FGD6, participant 23, aged 35)*

**Autonomy.** First, women spontaneously pointed out that education about risk factors can give rise to more awareness and prevention, because women may adopt risk-reducing lifestyle changes, such as quitting smoking. In general, women expressed a need for more information on risk factors and, specifically, more education on the relation between HPV (vaccination) and cervical cancer. The second subtheme is freedom of choice. The cervical cancer screening program, including the triage, could be adapted to a woman's preferences and personality. For example, some participants suggested that more anxious women could be offered a more intensive follow-up program. Moreover, women differed in whether they want to know their personal risk. Therefore, it was suggested to provide women the choice about whether they get to know their personal risk.

## Discussion

The aim of the present study was to gain insights on aspects that influence the acceptability of risk-based triage in cervical cancer screening. In total, five themes were found to describe the perceptions of the target group of the Dutch cervical cancer screening program. First, the most important aspect to women is an evidence-based screening program that is able to minimize cancer incidence and unnecessary referrals. Second, our focus group discussions revealed that aspects around personal information, such as the sensitivity of the questions, could influence support for risk-based triage. Third, cervical cancer screening should reduce fear and offer reassurance. Fourth, women preferred transparent, accessible communication around the screening program and suggest having face-to-face contact with a health care professional to talk about their personal cervical cancer risk. Fifth, higher autonomy with regards to prevention and screening behavior can increase acceptability of risk-based triage in the cervical cancer screening program.

Previous studies on acceptability of risk-based breast cancer screening [26–30] showed that women are positive about a more intense triage for high-risk women, and attitudes towards a less intense triage of women at a lower risk of cervical cancer were mixed. Our study extends this knowledge to cervical cancer screening: women appreciated a reduction of unnecessary referrals and check-up smears, but voiced concerns about potentially missing cancer.

In our study, most women were willing to provide sensitive information on sociodemographic and lifestyle risk factors. Previous studies also demonstrated that women are willing to provide information on potentially sensitive topics [16–18]. Fear of stigma as a result of having a higher cervical cancer risk was generally not considered an important factor. However, some women mentioned it might be an issue for other women, supporting previous literature that showed that women perceive stigma and discrimination around HPV (vaccination) and cervical cancer [31, 32].

Furthermore, although we did not include questions about prevention and it's not the focus of our study, some women spontaneously mentioned the effect of risk profiling in the triage of cervical cancer screening on their awareness of risk factors, their lifestyle, and their health. Knowledge on lifestyle factors that are associated with cervical cancer in the general population [16, 17, 33] and also among women who are hrHPV-positive [34, 35] can empower women to reduce their risk of this disease. However, the effects of most lifestyle risk factors may be limited, with the exception of smoking [16–18]. In addition to that, only providing information may be insufficient and additional strategies are needed to promote lifestyle change and maintain a healthy lifestyle [36, 37].

Before implementing risk-based triage in cervical cancer screening, it is important to take into account the aspects that are important to the population [38]. First, misclassification of

risk should be prevented, for example by providing a "don't know" or "don't want to answer" option to the questions, and by an appropriate frequency to assess modifiable risk factors such as smoking and marital status. Second, some practical issues may arise. For example, most women in our study indicated they would prefer a face-to-face conversation or phone call with their general practitioner or gynecologist to discuss their risk, especially when they are found to be at a higher risk of cervical cancer. However, this may place a larger burden on health care professionals. To give another example, a tension between a clinically optimal triage scenario and a preferred triage scenario can be expected when low-risk women experience a fear of having cancer. To mitigate these issues, special emphasis on communication of risk profiling and individual risk scores to women is essential. As already suggested in our study, using pictures in health communication can enhance comprehension, especially in low-literate individuals [39–41].

The current study is, to our knowledge, the first qualitative study on risk profiling in the triage of cervical cancer screening, and provides novel, in-depth insights in the perspectives of women. Another strength of the current study was the diverse sample with regard to age, education level, and cervical cancer screening experiences. We included both women who had participated in the cervical cancer screening program, and women who had never participated. Moreover, some women had experience with being tested positive for hrHPV, being invited for a check-up smear, or being referred to the gynecologist, while others had only received negative screening results. One potential limitation is self-selection bias towards individuals who are willing to share personal information. As a result of this bias, the perspectives of women who are less willing to share information, may be underrepresented in the current study. This point was also raised by some of the participants. However, some women in our study mentioned they would have difficulties sharing sensitive information, indicating our study succeeded in capturing a broad range of perspectives. Another potential limitation is that smallest FGD size was two participants, as a result of several last-minute cancellations. However, the smaller size did not impede the exchange of views and ideas.

In future studies, more attention is needed on the perceptions and views of non-Dutch-speaking women and of women who are harder to reach, including women who do not intend to participate in the current cervical cancer screening program. Furthermore, our findings could serve as a starting point for further quantitative studies on the perspectives of the target group. For example, a questionnaire study could investigate the overall acceptability of risk-based triage in the general population, and could examine the importance of the aspects identified in this study in a larger and representative sample of women. Moreover, quantitative studies could investigate differences between sociodemographic groups. Although in the Netherlands the first hrHPV vaccinated women will enter the program in 2023, studies on risk-profiling of HPV-positive women will remain relevant because of the relatively low (50–60%) hrHPV vaccination uptake [42] and because women who did not have the opportunity to take the HPV-vaccination (born before 1993) will continue to be invited for cervical cancer screening until 2052 [43]. HPV vaccination status will be an important risk factor to include in risk profiling.

To conclude, based on the in-depth insights in the perspectives of women, several challenges regarding the development and implementation of risk-based triage need attention. Broad support of risk-based triage will require, for example, an appropriate way of dealing with sensitive topics and a transparent communication strategy that provides reassurance and is also feasible. These points of attention do not only apply to cervical cancer screening, but also to other cancer screening programs, that will most likely move towards a more personalized approach in the next decades.

## Supporting information

**S1 Table. COREQ (COnsolidated criteria for REporting Qualitative research) checklist.**
(DOCX)

**S2 Table. Topic guide.**
(DOCX)

**S3 Table. Demographic and cervical cancer screening characteristics per participant.**
(DOCX)

**S4 Table. Example quotes per (sub)theme.**
(DOCX)

## Acknowledgments

The authors gratefully acknowledge the contributions of the Radboudumc General Practice Network to this study and wish to thank all participants for their input.

## Author Contributions

**Conceptualization:** Sharell Bas, Erik W. M. A. Bischoff, Ruud L. M. Bekkers, Inge M. C. M. de Kok, Willem J. G. Melchers, Albert G. Siebers, Daniëlle van der Waal, Mireille J. M. Broeders.

**Data curation:** Sharell Bas.

**Formal analysis:** Sharell Bas, Jasmijn Sijben, Daniëlle van der Waal.

**Funding acquisition:** Daniëlle van der Waal, Mireille J. M. Broeders.

**Investigation:** Sharell Bas, Jasmijn Sijben, Daniëlle van der Waal.

**Methodology:** Sharell Bas, Erik W. M. A. Bischoff, Ruud L. M. Bekkers, Inge M. C. M. de Kok, Willem J. G. Melchers, Albert G. Siebers, Daniëlle van der Waal, Mireille J. M. Broeders.

**Project administration:** Sharell Bas.

**Resources:** Erik W. M. A. Bischoff.

**Supervision:** Daniëlle van der Waal, Mireille J. M. Broeders.

**Writing – original draft:** Sharell Bas.

**Writing – review & editing:** Jasmijn Sijben, Erik W. M. A. Bischoff, Ruud L. M. Bekkers, Inge M. C. M. de Kok, Willem J. G. Melchers, Albert G. Siebers, Daniëlle van der Waal, Mireille J. M. Broeders.

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
