## [Decision Letter · Decision Letter 0]

22 May 2023

PONE-D-23-06691Acceptability of risk-based triage in cervical cancer screening: a focus group studyPLOS ONE

Dear Dr. Broeders,

Thank you for submitting your manuscript to PLOS ONE. After careful consideration, we feel that it has merit but does not fully meet PLOS ONE’s publication criteria as it currently stands. Therefore, we invite you to submit a revised version of the manuscript that addresses the points raised during the review process.

ACADEMIC EDITOR: Please review the paper for grammar and spelling errors. Also, respond to all comments raised by the two reviewers. 

We look forward to receiving your revised manuscript.

Kind regards,

Joseph Adu, MSc., Mphil

Academic Editor

PLOS ONE

Reviewers' comments:

Reviewer's Responses to Questions

**Comments to the Author**

1. Is the manuscript technically sound, and do the data support the conclusions?

Reviewer #1: Yes

Reviewer #2: Yes

2. Has the statistical analysis been performed appropriately and rigorously? 

Reviewer #1: Yes

Reviewer #2: Yes

3. Have the authors made all data underlying the findings in their manuscript fully available?

Reviewer #1: No

Reviewer #2: Yes

4. Is the manuscript presented in an intelligible fashion and written in standard English?

Reviewer #1: Yes

Reviewer #2: Yes

5. Review Comments to the Author

Reviewer #1: I found the manuscript technically sound and the findings support the conclusion of the research. Regarding full data availability the authors indicated restrictions would apply to full availability of the research data. The manuscript is written in standard English, however, some sentences highlighted in my review comments need to be corrected. While I consider this paper valuable because it provides useful findings on the topic and deserves to be published. It would however be useful **for the authors** to improve some aspects of the article. Find below my comments that could improve this work.

Could the authors describe how they selected the four general practices for the study?Could you state the number of participants recruited for the study in the method section?In line 33 age group should be moved from the end of the sentence. The sentence should read “Therefore, it is essential to gain knowledge of the views of women (30-60 years) eligible for cervical cancer screening in The Netherlands”.In lines 51 and 52 could you highlight any one of the challenges with risk-based triage that needs attention?Years should be added after women in line 62.Lines 67 to 70 should read “As a consequence, the detection rate for 68 cervical intraepithelial neoplasia grade 2 or worse (CIN2+) increased from 999 to 1461 per 69 100,000 among women screened between 2014/2015 and 2017/2018. On the other hand, clinically irrelevant findings (≤CIN1) increased from 611 to 1487 per 100,000 among women screened”.I am not clear about how the sentence in lines 84 and 85 relates to the research topic, especially so because apart from this sentence nowhere in the paper have you suggested the use of machine learning in this intervention. Could you provide further clarification?In line 116 I suggest you re-write the sentence “An information e-mail, which provided brief study information and the researchers’ contact details, was sent to all female patients aged 30 to 60 years old who were registered in four general practices”  to read “An email which provided brief information about the study and the researchers’ contact details, were sent to all female patients aged 30 to 60 years old who were registered in four general practices”.Provide the other dichotomized category for educational level in line 130.In lines 155 and 156 the authors reported that in “the largest FGD, engagement varied because of several reasons, such as distractions at home”. How did these distraction impact effective participation and responses from participants?In line 357 the authors indicated that “one of the strengths of the study was the diverse sample with regard to age, education level, and cervical cancer screening experience”. However, the study excluded non-Dutch-speaking women. How has this exclusion impacted the results of this study?  

Reviewer #2: This paper should be published after a few edits and clarifications. The authors need to review the manuscript heavily for grammar. The below comments are specific observations that the authors need to revise before consideration for acceptance.

General comments: The paper must be reviewed for grammar and spelling errors, i.e. see line 73 “…women is, since July 2022, based on..”

Methods:

Study design:

Lines 103: The author should provide more details on the design. Stating that it is a qualitative focus group does not tell readers what the methodology entails.

Line 103-108: Informed consent and ethical approval should move to the ethics section.

Line 133: If your preferred age range is 30 and 60 years old women, the statement about maximizing heterogeneity of women participating may be faulty. The authors already have a set parameter that limits true heterogeneity based on age. I suggest removing age and focusing on educational level, socioeconomic status, and CCS experience.

Line 136: Interview guide: I understand the need to call it an interview guide, but in a focus group discussion, while a group interview, the interviewer is more a facilitator than an interviewer. This leads back to my comment clarifying exactly what a qualitative focus group study means.

Line 154: “…on average four participants per FGD (2-7 participants)…” FDGs should be a minimum of 3 participants.

Lines 160-162: “…and were transcribed verbatim. Information that could lead to identification of participants, such as name or place of residence, was not included in the transcripts. The transcripts were not returned to the participants.” This should go down to analysis.

Line 170: citation

Line 177: (re)named. Why is this in parenthesis?

Results:

Line 244: (FGD3, p8, aged 46) Unless there is a specific reason why you need this level of detail, remove the page number.

6. PLOS authors have the option to publish the peer review history of their article (what does this mean?). If published, this will include your full peer review and any attached files.

Reviewer #1: **Yes: **Eliasu Yakubu

Reviewer #2: **Yes: **Mary Ndu

---

## [Author Response · Author response to Decision Letter 0]

20 Jul 2023

We have addressed the comments of the two reviewers below. In addition we have uploaded two files with the response to both reviewers. 

REVIEWER 1

PLOS One review

This paper should be published after a few edits and clarifications. The authors need to review the manuscript heavily for grammar.

General comments: The paper must be reviewed for grammar and spelling errors, i.e. see line 73 “…women is, since July 2022, based on..”

Author’s response: We thank the reviewer for pointing this out. We have reviewed the paper for grammar and spelling errors. 

Methods:

Study design:

1. Lines 103: The author should provide more details on the design. Stating that it is a qualitative focus group does not tell readers what the methodology entails. 

Author’s response: Indeed, we agree that more details about the methodology could have been included. We rephrased the sentence (line 101-102) “This is a focus group study” to “Focus group discussions (FGDs) were performed to explore women’s perceptions of risk-based triage in cervical cancer screening”. 

2. Line 103-108: Informed consent and ethical approval should move to the ethics section.

Author’s response: We changed the heading “Study design” into “Study design and ethics approval” (line 100). 

3. Line 133: If your preferred age range is 30 and 60 years old women, the statement about maximizing heterogeneity of women participating may be faulty. The authors already have a set parameter that limits true heterogeneity based on age. I suggest removing age and focusing on educational level, socioeconomic status, and CCS experience.

Author’s response: We thank the reviewer for this comment. We agree that diversity of the sample, for example with regards to age, is important. In The Netherlands, women from 30 to 60 years old are invited to participate in cervical cancer screening. Since the goal of our study was to gain more knowledge of the views of women eligible for cervical cancer screening, we were thus limited to the age range of 30 to 60 years old. However, within this age range, we feel that it is still relevant to maximize heterogeneity. With regards to socioeconomic status: apart from education level, we did not have information about the socioeconomic status of the participants. Therefore, we prefer to keep the sentence as it was in our manuscript (line 135-137): “[…] we aimed to maximize the heterogeneity of the women participating in the FGDs with regards to age, education level and cervical cancer screening experiences.”. 

4. Line 136: Interview guide: I understand the need to call it an interview guide, but in a focus group discussion, while a group interview, the interviewer is more a facilitator than an interviewer. This leads back to my comment clarifying exactly what a qualitative focus group study means.

Author’s response: We changed the term “interview guide” into “topic guide” to avoid confusion about the type of qualitative study (e.g. line 139). 

5. Line 154: “…on average four participants per FGD (2-7 participants)…” FDGs should be a minimum of 3 participants.

Author’s response: We agree that, ideally, a FGD should be a minimum of three participants. This specific FGD had two participants because of several last-minute cancellations. However, the smaller size did not impede the exchange of views and ideas. This FGD was an interesting and engaging conversation between the two participants. This FGD did not stand out compared to the other FGDs in terms of the views and ideas that were mentioned. We are also hesitant to discard data, since these participants have invested their valuable time in this focus group discussion. Therefore, we decided to include this FGD in the analysis of the study. The other six FGDs had at least three participants. 

6. Lines 160-162: “…and were transcribed verbatim. Information that could lead to identification of participants, such as name or place of residence, was not included in the transcripts. The transcripts were not returned to the participants.” This should go down to analysis.

Author’s response: We thank the reviewer for this helpful suggestion. We have moved this to “Data analysis” (line 170-172). 

7. Line 170: citation

Author’s response: It is not completely clear to us whether the reviewer suggests to add a reference to this line, or suggests to rephrase the sentence because it would be a citation. References about reflexive thematic analysis can be found in the previous sentence (line 172-173). We are not aware of this sentence being a citation. We have rephrased the sentence “Reflexive thematic analysis highlights the active and creative role of the researcher.” into “In reflexive thematic analysis, the researchers have an active and creative role.” (line 173-174). 

8. Line 177: (re)named. Why is this in parenthesis?

Author’s response: We aimed to explain the process of a part of the data analysis. In some cases, we started the discussion with only a description of the theme but did not have a name for the theme yet, i.e. we named these themes during the discussion. In other cases, we already had a name for the theme, but we decided another name would be a bitter fit, i.e., we renamed these themes. We have revised this sentence (line 179-181) as follows: “In several discussions, SB, DvdW, and JS explored different perspectives on the data and further developed, defined, named, and, where appropriate renamed, the themes and subthemes (step 4-5).”.

Results: 

9. Line 244: (FGD3, p8, aged 46) Unless there is a specific reason why you need this level of detail, remove the page number.

Author’s response: We apologize for our lack of clarity; the “p” refers to the participant number. We provided the participant number so that the reader is able to look up participant characteristics (Table S3) related to the quotes. We have changed “p8” to “participant 8” and did the same for the other quotes in the main text of the manuscript.

REVIEWER 2

Review comments: PONE-D-23-06691 

Dear Editor, Thank you for giving me the opportunity to provide a peer review for this research article titled, “Acceptability of risk-based triage in cervical cancer screening: a focus group study”. The article explored the factors that influenced the acceptability of risk-based triage cervical cancer screening among women 30-60 years in The Netherlands. 

While I consider this paper valuable because it provides useful findings on the topic and should be accepted for publication after the following improvements have been made to the article. Find below my comments that could improve this work. 

1. Could the authors describe how they selected the four general practices for the study? 

Author’s response: We thank the reviewer for this helpful suggestion. We added information about the selection of the four general practices (line 117-120): “The general practices were selected because they are core practices in the Radboudumc General Practice Network and are familiar with participating in scientific research. The practices are located in different neighborhoods in the city of Nijmegen and cover a broad range of socioeconomic status and ethnic backgrounds.”. 

2. Could you state the number of participants recruited for the study in the method section? 

Author’s response: Indeed, the number of participants should have been included in the methods section. Therefore, we changed the sentence (line 156): “In total, seven FGDs were held with on average four participants per FGD (2-7 participants).” into “Seven FGDs were held with 28 participants in total (2-7 participants per FGD).”. 

3. In line 33 age group should be moved from the end of the sentence. The sentence should read “Therefore, it is essential to gain knowledge of the views of women (30-60 years) eligible for cervical cancer screening in The Netherlands”. 

Author’s response: Agree. The age group has been moved as suggested by the reviewer (line 32).

4. In lines 51 and 52 could you highlight any one of the challenges with risk-based triage that needs attention? 

Author’s response: We agree that this information would be valuable to the abstract. Therefore, we have decided to shorten the abstract in order to add (line 52-53): “These challenges include dealing with sensitive topics and a transparent communication strategy.” 

5. Years should be added after women in line 62. 

Author’s response: We have changed this line into “[…] for all women aged 30-60 years” (line 61). 

6. Lines 67 to 70 should read “As a consequence, the detection rate for 68 cervical intraepithelial neoplasia grade 2 or worse (CIN2+) increased from 999 to 1461 per 69 100,000 among women screened between 2014/2015 and 2017/2018. On the other hand, clinically irrelevant findings (≤CIN1) increased from 611 to 1487 per 100,000 among women screened”. 

Author’s response: We have incorporated this suggestion (line 66-70). 

7. I am not clear about how the sentence in lines 84 and 85 relates to the research topic, especially so because apart from this sentence nowhere in the paper have you suggested the use of machine learning in this intervention. Could you provide further clarification? 

Author’s response: We thank the reviewer for pointing this out. The sentence about machine learning was included to give an example of our statement that the population benefit-harm ratio of cervical cancer screening can be improved by an optimized risk-based scenario. However, we agree that this sentence does not fit perfectly in our introduction. We have now removed the part about machine learning to emphasize the results rather than the methods (line 83-84): “A recent study has indeed shown that accurate risk classification can be improved to develop a personalized cervical cancer screening program (19).”

8. In line 116 I suggest you re-write the sentence “An information e-mail, which provided brief study information and the researchers’ contact details, was sent to all female patients aged 30 to 60 years old who were registered in four general practices” to read “An email which provided brief information about the study and the researchers’ contact details, were sent to all female patients aged 30 to 60 years old who were registered in four general practices”. 

Author’s response : We thank the reviewer for these helpful suggestions. We have revised these lines (line 115-117).

9. Provide the other dichotomized category for educational level in line 130. 

Author’s response: We agree that both education level categories should be further explained. Therefore, we have changed the part about educational level to add more information about the lower-intermediate category. The original sentence was “Education level was dichotomized, and a higher education level was defined as having completed university or university of applied sciences, which is consistent with the classification of Statistics Netherlands (22).”. The new sentences read as follows (line 130-134): “Education level was dichotomized consistent with the classification of Statistics Netherlands (22). A higher education level was defined as having completed university or university of applied sciences, and a lower-intermediate education level includes primary education, prevocational secondary education, senior general secondary education, pre-university education, and senior secondary vocational education.”

10. In lines 155 and 156 the authors reported that in “the largest FGD, engagement varied because of several reasons, such as distractions at home”. How did these distraction impact effective participation and responses from participants? 

Author’s response: We understand that this sentence raises the question whether/how the variable engagement impacted the responses from the participants. Therefore, we have added the following (line 159-160): “However, the (partial) absence of some participants did not negatively impact the active discussion between the other participants.”. 

11. In line 357 the authors indicated that “one of the strengths of the study was the diverse sample with regard to age, education level, and cervical cancer screening experience”. However, the study excluded non-Dutch-speaking women. How has this exclusion impacted the results of this study?

Author’s response: We decided to only include Dutch-speaking women in order to conduct the FGDs in Dutch. English-speaking FGDs would exclude women who are not proficient in English, and we did not aim to conduct an FGD with only non-Dutch-speaking women. We agree with the reviewer that a diverse sample is important, and we do agree that we might have missed some perspectives by excluding this group. We hope future research will study the perspectives of a broader group of women, and we added non-Dutch-speaking women to our paragraph about future research (line 375-377): “In future studies, more attention is needed on the perceptions and views of non-Dutch-speaking women and of women who are harder to reach, including women who do not intend to participate in the current cervical cancer screening program.”

---

## [Editor Report · Decision Letter 1]

24 Jul 2023

Acceptability of risk-based triage in cervical cancer screening: a focus group study

PONE-D-23-06691R1

Dear Dr. Broeders,

We’re pleased to inform you that your manuscript has been judged scientifically suitable for publication and will be formally accepted for publication once it meets all outstanding technical requirements.

Kind regards,

Joseph Adu, PhD, MSc., Mphil

Academic Editor

PLOS ONE

Additional Editor Comments (optional):

Thank you for submitting your manuscript to PlosOne Global Health Journal. You have carefully addressed the reviewers' comments.

Your manuscript is accepted without futher revisions. You are required to format your references in accordance with journal style.

Thank you once again for the opportunity to review your article.

Joseph Adu, PhD
---

## [Editor Report · Acceptance letter]

8 Aug 2023

PONE-D-23-06691R1 

Acceptability of risk-based triage in cervical cancer screening: a focus group study 

Dear Dr. Broeders:

I'm pleased to inform you that your manuscript has been deemed suitable for publication in PLOS ONE. Congratulations! Your manuscript is now with our production department. 

Kind regards, 

on behalf of

Dr Joseph Adu 

Academic Editor

PLOS ONE